# Understanding and Awareness of Dementia in the Pakistani-Origin Community of Stoke-on-Trent, UK: A Scenario-Based Interview Study

**DOI:** 10.3390/healthcare12020251

**Published:** 2024-01-19

**Authors:** Nargis Nazir, Peter Kevern

**Affiliations:** School of Health, Sciences and Wellbeing, Staffordshire University, Stoke-on-Trent ST4 2DE, UK

**Keywords:** minority, ethnic, dementia, Pakistani community, Pothwari

## Abstract

The aim of this project was to explore the understanding and awareness of dementia in the Pakistani-origin community in a deprived urban region of the UK. The study was unique in accessing and interviewing Pothwari speakers, some of whom could not read or understand spoken English. Data generated from an earlier study were used to construct five scenarios, which were used as the basis for face-to-face semi-structured interviews with 11 male and female participants from the Pakistani-origin community spanning two generations. Braun and Clarke’s six phases of thematic analysis were used to analyse the data to answer the research questions. Themes constructed from these interviews indicated a lack of awareness and understanding of dementia, a range of attitudes and assumptions, reluctance to seek external support, and a significant role for the cultural background in shaping the individuals’ responses. The study found that poor understanding, cultural differences, and language issues presented barriers to accessing services in the British Pakistani community, particularly among those who had been born in Pakistan and/or spoke Pothwari in preference to English. Services and information may need to be offered by Pothwari speakers in order to reach this neglected sector of the population.

## 1. Introduction

Dementia is a growing global challenge that affects millions of individuals worldwide [1] and is most common in people 65 years of age or older [2]. In the UK, an estimated 944,000 people are living with dementia, and this is predicted to increase to 1.6 million by 2030 [3]. Black, Asian, and Minority Ethnic (BAME) communities are ageing alongside the White British population [4], and the increasing numbers of people over 65 in many BAME communities means that the proportion living with dementia will rise at a rate faster than for the population as a whole. There are currently 25,000 people living with dementia from the BAME groups in the UK. This is around 3% of the UK’s total prevalence, and this figure is estimated to grow to 50,000 by 2026 [3]. 

The Pakistani-origin community is the second largest non-white community in the UK after the Indian-origin community, and currently comprises 1.6 million individuals or about 2.7% of the total population [5]. Data from the 2021 census show that the Pakistani community is the second most numerous group in Stoke-on-Trent after white British (85.6%). There is a population of 6000 (2.3%) British-born Pakistanis living in Stoke-on-Trent, but there are no data for the individuals who were born in Pakistan, so the total size of the Pakistani-origin community is unknown [6]. 

The diagnosis rate for dementia in England in 2022 is 62%, which means that more than a third of cases are currently undiagnosed [3]. A disproportionate number of these are thought to be concentrated in BAME communities, where diagnosis tends to happen later in the disease trajectory. Some BAME communities may have an increased risk of dementia because of high rates of high blood pressure, stroke, cardiovascular disease, and diabetes [7], but even though BAME communities are probably at greater risk of developing dementia, there is still little understanding of dementia in some of these communities [8,9]. Symptoms of dementia may be mistakenly associated with “getting old” or mental illness by carers in some BAME groups [10,11]. Also, the stigma surrounding mental health acts as a barrier to accessing dementia-related services [12,13]. 

All of these factors are likely to be contributing to the pattern of late diagnosis and poor access to support services in some BAME communities. There is evidence that individuals with dementia from BAME communities approach dementia services later when their dementia is more advanced compared to their White British counterparts [10], and this results in seeking help from dementia services only once dementia is in its severe stage [14]. However, there has been relatively little research into specific minority communities in the UK setting, which is itself likely to be contributing to their poorer outcomes [15]. 

As well as a general lack of knowledge and awareness, there are specific cultural and contextual reasons why dementia may be underdiagnosed and underreported in the Pakistani-origin community. In Muslim Pakistani society, family status is very important, and it can be affected by *Izzat* (honour) and *Sharam* (shame). For example, the family *Izzat* could be lost through *Sharam* if a family reveals that a family member has dementia [16]. A diagnosis of dementia in a family member can have a seriously damaging effect on children’s marriage proposals, and the belief in the BAME population that mental illnesses cannot be cured and are genetic can reduce the chances of young adults getting married and may possibly lead to divorce [17]. 

The purpose of this research is to gain a better understanding of the level of knowledge and awareness of dementia in the Pakistani-origin community, along with insight into what barriers impede timely diagnosis and access to services. This will contribute to initiatives to improve community understanding of dementia and the services available, and so may eventually improve timely diagnosis and support, while reducing the effects and burden of stigma for family carers and increasing timely access to support services. 

## 2. Materials and Methods

### 2.1. Methods

The design of the present study drew on responses to questions gathered in a previous interview-based study of Pakistani Muslim family carers of people living with dementia living in the UK [18]. Responses were used to construct five scenarios that encapsulated common perceptions reported in the study (Figure 1), and these (with supplementary probing questions) formed the basis of an interview schedule used with participants.

### 2.2. Participants

A sample of 11 male and female participants, in an age range between 20 and 78, were recruited for this study through a snowballing process using personal contacts and neighbourhood networks. All the recruitment was undertaken from January to June 2023. 

Seven female participants were born in Pakistan, and two male and two female participants were born in the United Kingdom (UK). This was the maximum number that could be obtained from the community and reaches the threshold for the recommended sample size for thematic analysis of in-depth interviews [19,20]. The seven participants who were born in Pakistan could not communicate in English, so the researcher conducted the interviews in Pothwari (Pothwari is a regional language spoken in Pakistan). The interviews with the four participants born in the UK were conducted in English.

Once the potential participant had agreed to participate in the project, the researcher gave them the information sheet and the consent form (in English and Urdu) at their house. The information sheet and consent form were explained in Pothwari by the researcher to all the participants who could not read English or Urdu. Once consent had been secured, an interview date was arranged, and continuing consent was affirmed verbally at the start of each interview (Note: Urdu is the national language of Pakistan and Pothwari is a dialect of Punjabi. Pothwari is the regional language of Pakistan where the participants originated from. Pothwari does not have a written form and can only be written in Urdu script). 

### 2.3. Data Collection

Interviews took place in English and Pothwari between 16 January and 2 June 2023. Participants were given the scenarios (written in English and Urdu, but also read out in Pothwari for those who were unable to read either language) by the principal researcher on the interview day and were asked questions on the basis of the scenarios with additional prompts. Participants unable to read or understand English or Urdu were briefed on the scenarios in Pothwari and were then asked questions in Pothwari.

Audio recordings of the Pothwari interviews were translated and transcribed and then translated into English by the principal researcher. All the audio recordings of the English interviews were also transcribed manually. The transcripts were then used for the thematic analysis.

### 2.4. Data Analysis

Braun and Clarke’s six phases of thematic analysis were used to analyse the data to answer the research questions [19]. The first phase of data analysis (familiarization with data) was undertaken by the researcher, where each of the transcripts was read several times. Then, each of the scripts was highlighted if it had a clear reference to the research question. The next phases of generating the initial codes and searching for themes were undertaken, and then the initial themes were developed from long lists of codes. The themes were reviewed to make sure that there were enough data to support them. Finally, the report was produced after the themes were further defined and refined, and each theme was given a name.

## 3. Results

Analysis of the qualitative data generated four themes: (1) awareness and understanding of dementia, (2) attitude and assumptions towards dementia, (3) support, and (4) cultural norms (see Table 1). Quotations in the account below are given a participant code as follows:PbP—Participant born in Pakistan (Pothwari-speaking).PbUK—Participant born in the UK (English-speaking).

### 3.1. Awareness and Understanding

#### 3.1.1. Lack of Knowledge and Understanding

Generally, understanding of dementia among the Pakistani-origin community was relatively low; four of the interviewees born in Pakistan only became aware of dementia when one of their family members or a relative was diagnosed. Some of the participants were unfamiliar with the dementia symptoms; one participant reported that people in the Pakistani-origin community only approach outsiders for help when dementia is in its advanced stage. Some examples are listed below:

“*In regard to Pakistani community, I don’t really know anyone who’s got dementia. I don’t know what it is but I feel like dementia is not well really known, it’s like because lots of people talk about how forgetful they are, and to where point it becomes very serious no one really diagnosed dementia, especially I just come back from Pakistan and like they always say he passed away but his memory wasn’t there*.”(PbUK 10)

“*To be honest I just recently like a year and a half ago came to know about this illness when my brother said my father might have this. Prior to this, I have never heard this word. Never knew that this illness occurred … because in our community there is no awareness about this illness*.”(PbP 1)

“*So I know about this story about this dementia is from my sister-in-law maybe under a year or eight-nine months she told us about her dad’s suffering with this disease that’s how I know. … a lot of people not just in Pakistan here is well they don’t know about this disease*.”(PbP 6)

However, some who were born and raised in the UK had a slightly better understanding of dementia:

“*Think by the reading of that [scenario] it sounds like she’s getting quite early signs of dementia I am going to assume*.”(PbUK 10)

“*I think she has dementia. Because first it started off with her being forgetful and now it’s deteriorated it’s gone worst*.”(PbUK 11)

#### 3.1.2. Misrecognising Behaviour

Participants also indicated that Pakistani people do not recognise the symptoms of dementia and this may be one reason why families delay seeking for help until dementia is in its advanced stage:

“*Forgetting about the keys, we think that perhaps it’s normal that we put something somewhere and forget where we put it. when someone starts not to recognise their family members then you start to think but, in our community, we don’t take them to the doctors to get any treatment. We don’t do that. When things get very serious, when they start to forget what they did a little while ago, what they ate for their meals then we think it’s something very serious*.”(PbP 1)

“*First, when it all started, we could not figure out what was wrong with him. At night when he used to go to sleep, he could hear voices and he used to say that someone walks around our house…*”(PbP 5)

The symptoms of dementia are often medically unnamed in the daily lives of patients, their families and those in contact with people affected by dementia. Signs of ‘strange behaviour’ may be observed, but usually are not identified as indicators of dementia. The participants of the study were asked if they would recognise the symptoms of dementia:

“*I mean the main one is probably forgetful but that’s so common, but I probably say no I would not recognise because let’s say we were talking about other scenario when someone’s been aggressive that could be the case of that’s probably their personality. So my impression is this is how they are like in real life. So I probably wouldn’t recognise it*.”(PbUK 10)

### 3.2. Attitudes, Assumptions, and Perceptions towards Dementia

The accepted understanding of dementia in western discourse is that the term refers to a condition with a range of causes that is treated as having an organic basis: that is, it is not a normal part of the ageing process, or a form of mental illness. By contrast, participants reported a number of attitudes and perceptions of dementia which reflected the relatively poor understanding within the community at large:

“*Mrs. N is busy with her work, but some people may think that Mrs. N’s mum is doing all this on purpose to get attention because she has nothing else to do. She is fed on time, her laundry is done on time, and she is bathed on time, but she still bothers me…*”(PbP 1)

“*They talk nonsense. Like we think first then talk they don’t do that. You can tell the difference between a sensible person and a mindless person, that this person is sensible, and this person is mindless*.”(PbP 4)

“*In Pakistan, people can’t afford to get treatment. Occasionally people there are roaming like a madman*.”(PbP 4)

“*Myself, I tend to look at medical things anyway. If it is a dead creepy one, I have to glance at that*.”(PbUK 9)

“*They will call him as a mental, they will say he has mental health issues. They will register him for that straight away. … but a lot of people not just in Pakistan here as well they don’t know about this disease*.”(PbP 6)

“*I just come back from Pakistan and like they always say he passed away but his memory wasn’t there. And their assumption is old age*.”(PbUK 10)

### 3.3. Support

#### 3.3.1. The Primary Role of Family

People in the Pakistani-origin community prefer a family member with dementia to be looked after by family at home. Most participants said if they needed any caring support (in relation to dementia care), they would approach their family first, and seeking support from dementia care services was their last option. Some participants see carers as outsiders and will not seek help from them:

“*I will have to leave my job and take care of them or ask my siblings to look after them while I am at work. Outsiders are not going to help*.”(PbP 2)

“*I will go to my sisters and brothers for help and if they don’t help me then I will seek help from social services*.”(PbP 3)

“*First I would say to my immediate family and if there is no other option, then I will contact my doctor and ask for someone who can help me at night*.”(PbP 5)

“*If I was in that position I’d try to, I’d try to … get more support from my family. Instead of putting her in a home or something else*.”(PbUK 11)

#### 3.3.2. Rejection of External Support

Some people who have a family member with dementia knew that support was available for people living with dementia and their families but did not use help, believing that the family was there to care for people affected by dementia. Furthermore, not everyone with dementia wants support from support services:

“*Yes, the dementia specialist asked us if we need any support, but we said no we have all the family, and I am home all the time*.”(PbP 5)

“*Family was the big support then and sometimes hospital, but you know … my great grandma, she wasn’t really much into support from hospital. She just wanted family around her*.”(PbUK 11)

### 3.4. Cultural Norms

#### 3.4.1. Knowing Where to Look

In the Pakistani-origin community, a General Practitioner (GP) is perceived as the first line of help and support for people affected by brain function change. Four of the participants presented with a scenario in which a family carer needs to seek help with a person living with dementia indicated this as their immediate response (PbP 1, PbP 7, PbUK 9, PbUk 10): 

“*First she needs to go to see her doctor (GP) for advice and get the support where her doctor suggests for example her mum needs counselling or refer her mum to a mental health specialist*.”(PbP 1)

Apart from this response, six of the eleven participants reported they had no knowledge of the support available for people living with dementia and their families, and this seemed to be equally the case for those born in the UK and those born in Pakistan:

“*Don’t know if there was support available. I thought families just do it themselves*.”(PbUK 8)

#### 3.4.2. Cultural Differences

In this study, it was found that participants were deterred from making use of support services by perceived cultural differences: 

“*In our community, it will be hard that our elders will agree to go to the dementia daycare centre and stay there on their own in a different culture. If there are people just from our community then they will probably agree to go, because if they are just with English people, they will not understand the language, and they will not like the food*.”(PbP 1)

“*I don’t think they will. I think because Pakistani community they quite, they need to be if it’s specifically for the Pakistani community. I don’t think they will be comfortable with someone else So obviously they will start speaking their language and if there no one there who can communicate with them then obviously the family members will be worried*.”(PbUK 10)

Nevertheless, the beneficial effect of dementia daycare centres was appreciated by one of the study participants, who suggested that the daycare centres would provide respite for carers:

“*I think they would approach because it can be difficult on a family. But I think maybe if they were Pakistani too, it will be more likely they can understand the patient*.”(PbUK 8)

#### 3.4.3. Willingness to Use Residential Services

Some family members of individuals with dementia are willing to seek help from carers; if necessary, they are ready to place their relatives in a care institution.

“*Family can’t stay because everyone has their own families. You can hire someone and pay them, or the council can help you…*”(PbP 4)

“*I would ring GP and ask for the social worker. To see if they could help with the carers or potentially move to a care home*.”(PbUK 8)

## 4. Discussion

The findings from the first theme indicate that, generally, participants born in Pakistan are not aware of dementia, although those who become aware when a family member or relative was diagnosed show an increased awareness, despite a limited understanding. Participants born in the UK have a slightly better understanding of dementia and of the likely range of symptoms. The findings suggest that educating this community will increase awareness, understanding and recognition of dementia, particularly among Pothwari-speakers.

To summarise the second theme, there is clearly a need to reduce stigma and misunderstandings of dementia in the Pakistani community as a whole. A lack of understanding about dementia can lead to various misconceptions, resulting in the perpetuation of stigma, but also indirectly to failures in family-based care and a delay in involving statutory service providers. 

The evidence from the third theme bears out findings from previous studies that suggests that Pakistani culture has a strong impact on willingness to look after a family member with dementia at home. The expectation of the community is that a person living with dementia will be cared for by their family, and that caring for an elderly family member is seen as a moral duty. 

There is now widespread acknowledgement that a variety of barriers inhibit help-seeking by and support for BAME people living with dementia [21]. Where outside help was to be sought, there was little awareness of services available but a strong tendency to see the GP as the first line of help and support. 

Also, participants from both generations highlighted barriers which will prevent older people from the Pakistani community going to dementia daycare centres, particularly if they cannot speak English. The question of whether participants would be willing to use residential facilities was not directly asked in the interviews, and in most cases was not raised by the participants. Placing close family members in a care home is difficult, but sometimes, it is the only reasonable option [22]. There remains a strong resistance within the community to using residential facilities, but this may be starting to change with changing demographics in the community. 

Access to both residential and daycare services was impeded by cultural concerns and, in the case of Pothwari speakers, the language barrier. There is a case, therefore, for multilingual daycare provision that includes staff who can speak and understand Pothwari. This might encourage the individual with dementia and their families from the Pakistani community to use the dementia day-care services and feel comfortable doing so.

The lack of understanding and awareness of dementia identified in this study has been reported in previous studies. Jeraj and Butt (2018) found that awareness of dementia in BAME communities is limited [9]. Due to a lack of understanding and awareness of dementia, participants born in Pakistan only became aware of dementia when a family member or a relative was diagnosed. The present study also found that some family members in the Pakistani community may think that the individual is acting to seek the family’s attention [23]. Additionally, this study identified that there is a lack of knowledge about dementia symptoms amongst individuals born in Pakistan and in the UK [11], and this may contribute to a situation in which help from services is only sought when dementia is in an advanced stage [14].

While in European countries, dementia is recognised as a disease associated with behavioural change and memory loss, affecting people’s quality of life [1], the evidence from this and previous studies is that some members of the Pakistani-origin community see people affected by dementia as a “madman” and “mindless” [24]. Also, dementia is seen as something “creepy” by some in the community [25] and is viewed as a mental illness [26]. Each of these studies point out that there is a significant discrepancy between the level of knowledge and understanding of dementia in the ethnic Pakistani community when compared to the majority-white population in the UK [24,25,26]. Hence, there is a need for more community-level education about dementia in order to facilitate compassionate care and support.

The results from this study suggest that many people in the Pakistani-origin community believe that it is their moral duty to care for their loved ones with dementia [27]. The study also found, as with previous studies, that there is a strong Pakistani tradition to “look after their own” and only allow close relatives to provide support [16,17]. This study further suggests that the parents are looked after in their own homes and care provided at home is considered superior to that provided in care homes [7,16]. However, two participants reported in this study that they would use the mainstream services which provide caring, support, and residential services, and this may be the first early indication that attitudes are shifting.

Based on this study, it has been established that the GP is seen as the first line of help and support for people from the Pakistani society [16]. Balarajan, Yuen, and Raleigh (1989), similarly, found that the number of Pakistani individuals utilising GP services was high [28]. However, studies have shown that GPs may be slow to recognize and diagnose dementia for a range of reasons, including assumptions that are made about the role of the family in minority-ethnic communities, as well as language and cultural barriers [29]. Consequently, this community may continue to suffer from late diagnosis without more adequate signposting to other sources of support and information: many people in the Pakistani-origin community are unaware of the support and services available for people living with dementia and their families [13,25]. 

Some participants in this study reported that they were aware of the services available. This study also identified that there are barriers to accessing dementia care services, and both Pakistan-born and UK-born participants reported that people with dementia will not use dementia daycare services because of cultural considerations and language difficulties [18]. Thus, training and education within primary and secondary care services are essential to reduce barriers to access.

### Limitations

The sample size for this study was comparatively small with only 11 participants, which reflected the difficulty accessing this community as a whole [18]. Also, some interviews were short because some Pothwari-speakers were unused to the interview format and sought to give a single “right” answer. Currently, no platform converts the Pothwari language into English, so translating the Pothwari interviews into English is time-consuming, with differences in sentence structure and vocabulary leading to potential mistranslation. It is a limitation of this study that the transcripts were not checked by a third party. However, the strength of this study was that, by including Pothwari speakers, it was possible to gather data spanning two generations in the Pakistani community and include an older generation whose perceptions have not so far been explored. 

## 5. Conclusions

As the ethnic Pakistani population is ageing alongside the white British population in the UK, dementia is also becoming increasingly common in that community. Population health needs are starting to converge: dementia in the Pakistani-origin community will become increasingly common and a burden on the NHS. The findings from this study suggest that the understanding of dementia in the Pakistani-origin community living in Stoke-on-Trent is limited, with some evidence of a negative impact on the individual with dementia and their families. Therefore, there is an urgent need for dementia awareness campaigns to improve the understanding of dementia in the community to reduce stigma, improve access to support, and improve the quality of life for people living with dementia and their carers.

Since the study found that the GP is considered the first line of help and support, this should be the site for the initial education campaign; printed information should be displayed in GP surgeries in English and Urdu to make people aware of dementia support services. Among the factors that prevent people from accessing secondary care services, participants from both generations reported that cultural and language barriers prevent people with dementia from the ethnic Pakistani community accessing dementia daycare services. This indicates that there is an urgent need for health care services to gain knowledge and experiences about the Pakistani cultural background and employ bilingual staff in mainstream daycare centres who can communicate with people with dementia from that community. The findings from this study also suggest that it is very common in Pakistani culture for families to look after a family member with dementia at home, because families think it is their moral duty to look after an elderly family member. Providing care for a family member with dementia is a long-term commitment and can have a negative impact on the carer’s health. Hence, dementia support services need to reach out to the Pakistani-origin community to provide the individual with dementia and the carer (or carers) the right support they need.

## Figures and Tables

**Figure 1 healthcare-12-00251-f001:**
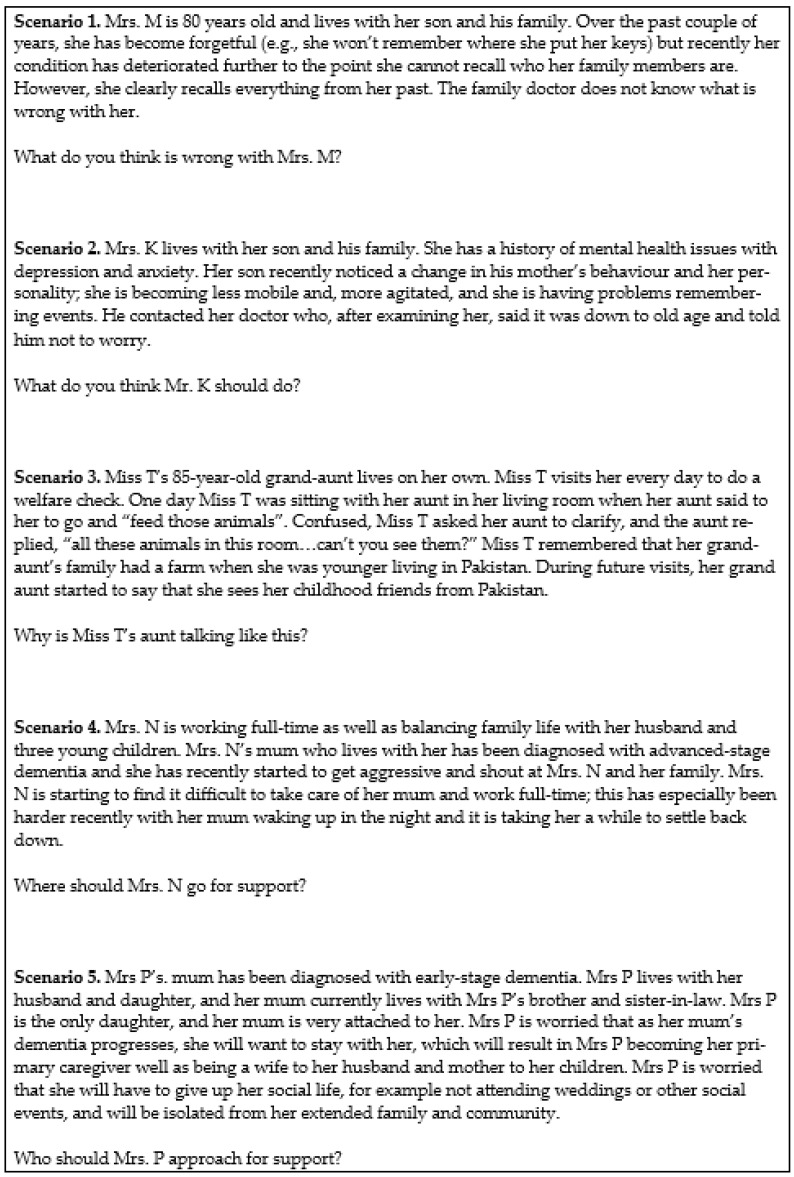
Scenarios derived from previous study.

**Table 1 healthcare-12-00251-t001:** Themes and subthemes.

Theme	Subthemes
1. Awareness and understanding	Lack of knowledge and understanding
	Misrecognising behaviour
2. Attitudes, assumptions, and perceptions towards dementia	
3. Support	Primary role of family
	Rejection of external support
4. Cultural norms	Knowing where to look
	Cultural differences
	Willingness to use residential services

## Data Availability

Interview data may be obtained by application to the authors of this paper.

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
