# Peer review of "Understanding and Awareness of Dementia in the Pakistani-Origin Community of Stoke-on-Trent, UK: A Scenario-Based Interview Study"

_healthcare, 2024, doi:10.3390/healthcare12020251_

Round 1

Reviewer 1 Report

Comments and Suggestions for Authors

I appreciate the opportunity to review the article titled "Understanding and Awareness of Dementia in the Pakistani-Origin Community of Stoke-on-Trent." The work is undoubtedly a significant contribution to the field. However, I have identified some points that require consideration and revision:

1. Lines 49-50: The citation (Mackenzie, 2006) appears dated. Consider updating it if more recent sources are available.

2. Line 141: The statement seems to contradict the author's claim regarding language barriers in the study limitations. Clarification or revision is needed either in this section or the limitation section.

3. Lines 158-159: Was there another person involved in cross-checking the translated transcriptions? This information should be provided for transparency.

4. Results Section (Lines 229-230 and more): The author should focus solely on what the study revealed, refraining from incorporating evidence from other studies. Save such discussions for the introduction or discussion sections.

5. Lines 248-251 (and others in results): The author's interpretation should not be included in the results. Stick to presenting participants' statements without interpretation and previous evidence.

6. Lines 271-274: Similar to the above comment, avoid incorporating author interpretations in the results section.

7. Discussion (Lines 447-448): The statement may need rewriting as it aligns with established findings from previous studies, making it less novel.

8. Lines 470-471: The citation is outdated considering the current study was conducted in 2023.

9. Sentence at an unspecified location: "Hence, dementia awareness needs to be raised, so positive steps can be taken to support individuals with dementia and their carers in the Pakistani community." Consider rewriting for clarity and a more critical tone.

10. Lines 482-483: This content should be included in the findings section.

11. Lines 502-503: The information about the first author being a Pothwari speaker should be reconciled with the earlier mention in the limitations section.

12. Line 517: Avoid using "so"; consider replacing it with "therefore" or similar for improved writing.

13. Lines 522-524: The sentence is lengthy; please break it down for better readability.

14. Lines 531-534: This content seems more suitable for the results section than the conclusion.

I believe addressing these points will enhance the overall quality and clarity of the article. Thank you for considering my feedback.

Best regards,

Comments on the Quality of English Language

As above

Author Response

We are grateful to the reviewer for their helpful comments. Our responses are as below:

  1. This reference has been replaced with one from 2017
  2. The apparent contradiction was because of some clumsy wording in lines 526-8, which has now been corrected
  3. The translations of interview materials and Urdu transcription were translated in collaboration with Mohamed Akram, who has now been named in the Acknowledgements. However, interview transcripts were not checked by a third party and this has been added to the limitations
  4. References have been deleted from the results section
  5. Interpretations have been removed from the Results section
  6. As above
  7. We were not able to locate this reference, as 447-8 is not in the Discussion section
  8. We were not able to locate a citation in the lines given
  9. Thank you for this comment. The sentence has been rewritten
  10. Inserted in the results section (429)
  11. The limitations section has been revised to resolve the apparent conflict
  12. Corrected
  13. Corrected
  14. This content is in the Results section, and is repeated in the Conclusion for the sake of clarity

Reviewer 2 Report

Comments and Suggestions for Authors

This paper is a rare study that analyzes the understanding and awareness of dementia among Pakistani-origin communities with language barriers. However, I think the following points require major revision.

1. Format: Unnecessary spaces and line breaks can be seen everywhere. It is also incorrect to have Table 1 divided into pages.

2. Abstract : The methods used for the analysis should be clearly described.

 3. Selection of subjects : It is not clear how the 11 participants, who ranged in age widely, were selected.

4. Result : It is necessary to clearly state how the four themes were extracted.

For each topic, you should summarize the points rather than list examples.

I think it is necessary to summarize the notable differences between PbP and PbUK.

Author Response

Thank you for your helpful comments. We have responded to each one as below

  1. Line breaks and formatting issues have been addressed
  2. The Abstract has been amended to include reference to Braun and Clarke's analytical process
  3. Additional information has been included on the recruitment process
  4. The process of extraction of themes from lists of codes is well established in Braun and Clarke's methodology and will be well known by those familiar with the field
  5. We have chosen to give examples under each theme because this is the established method of reporting results in qualitative research using this form of thematic analysis. We are concerned that if we removed these quotes the paper would have no validity in the eyes of the academic community
  6. Thank you for this suggestion. A few sentences summarizing the difference between those born in the UK and those born in Pakistan have been added at the beginning of the Discussion

Reviewer 3 Report

Comments and Suggestions for Authors

The authors present an interesting topic regarding awareness of the problem of dementia among the population of people of Pakistani origin in the UK. In the introduction, the authors suggest that the cultural background in this group, combined with the lack of sufficient substantive knowledge, may result in a lower tendency to diagnose dementia.

The authors refer to an earlier study on the Muslim population in the UK. The number of respondents has reached the threshold for the recommended sample size for thematic analysis of in-depth interviews. Braun and Clarke's six phases of thematic analysis were used to analyze the data.

The results section presents the conclusions from the research, which should be shown in the discussion. This section should present the results, which should be discussed later in the article, as should reference the literature. This is also not the place to make recommendations. The discussion should focus more on the unique difficulties present in this population, especially those identified during the study.

Expanding the discussion on the role of GPs in diagnosing dementia could be considered, which is crucial regardless of the population being examined. In addition to unique difficulties related to the study population, such as the language barrier, they also include insufficient awareness and willingness to diagnose cognitive disorders by GPs.

Author Response

We are grateful to the reviewer for their helpful comments. Our responses are summarised below.

1) "The results section presents the conclusions from the research, which should be shown in the discussion. . . " A clear separation has been established between the Results and discussion, and the discussion section has been rewritten to reflect this

2) "Expanding the discussion on the role of GPs. . .". This is not a central element of the research, but we agree that it needs more recognition. We have expanded our discussion of the point in lines 457-466

Reviewer 4 Report

Comments and Suggestions for Authors

This is a fine paper. Motivation, context, method and results are clearly presented. Qualitative studies like this one are the only way to reveal hidden nooks in society that cannot be reached by other social science methods. I have just a question, a very minor remark and a request.

Question: why are all the dementia patients in the scenario's female?

Remark: l. 271-72 "those who become aware when a family member or 271 relative was diagnosed are responding positively towards dementia" Not clear to me what is meant by 'positively' here.

Request: the paper would gain in strength if a comparison could be made, using other research and even if only indicative, with the attitudes of the white towards dementia and their knowledge of dementia and care options.

Author Response

We thank the reviewer for their positive and helpful comments. Regarding the specific points made:

1) The reviewer has misunderstood regarding the sex of the participants. They are not all female: "A sample of 11 male and female participants, age range between 20 to 78, were recruited for this study . . .Seven female participants were born in Pakistan, and two male and two female participants were born in the United Kingdom (UK)." (line 125-129)

2) Thank you for pointing out the unclear sentence. It has now been revised to read " . . . those who become aware when a family member or relative was diagnosed show an increased awareness, despite a limited understanding." (line 250-251)

3) The comparison with the attitudes of the majority white population is discussed in brief in lines 452-63, and we have added a sentence here to make the comparison explicit from the literature. However, since we did not include any white participants in our own study, we can only refer to the work of others for this information

Round 2

Reviewer 2 Report

Comments and Suggestions for Authors

I have never read a paper with unnecessary line breaks in the session. Also, I haven't read any recent qualitative studies that don't have clear criteria or path diagrams. I am not an expert in qualitative research, so I will decline to serve as a reviewer.

Author Response

We are sorry to hear that the reviewer has decided to withdraw from the process but thank them for their contribution to the first round of reviews.

Reviewer 3 Report

Comments and Suggestions for Authors

My comments have been adequately answered.